# Synthesis, Herbicidal Activity, Crop Safety and Soil Degradation of Pyrimidine- and Triazine-Substituted Chlorsulfuron Derivatives

**DOI:** 10.3390/molecules27072362

**Published:** 2022-04-06

**Authors:** Lei Wu, Yu-Cheng Gu, Yong-Hong Li, Sha Zhou, Zhong-Wen Wang, Zheng-Ming Li

**Affiliations:** 1State Key Laboratory of Elemento-Organic Chemistry, College of Chemistry, Nankai University, Tianjin 300071, China; wl836929871@163.com (L.W.); bioassay@nankai.edu.cn (Y.-H.L.); zhousha@nankai.edu.cn (S.Z.); 2Syngenta Jealott’s Hill International Research Centre, Bracknell RG42 6EY, Berkshire, UK; yucheng.gu@syngenta.com

**Keywords:** sulfonylurea herbicides, herbicidal activity, soil degradation, crop safety, DT_50_

## Abstract

Chlrosulfuron, a classical sulfonylurea herbicide that exhibits good safety for wheat but causes a certain degree of damage to subsequent corn in a wheat–corn rotation mode, has been suspended field application in China since 2014. Our previous study found that diethylamino-substituted chlorsulfuron derivatives accelerated the degradation rate in soil. In order to obtain sulfonylurea herbicides with good crop safety for both wheat and corn, while maintaining high herbicidal activities, a series of pyrimidine- and triazine-based diethylamino-substituted chlorsulfuron derivatives (**W102**–**W111**) were systematically evaluated. The structures of the synthesized compounds were confirmed with ^1^H NMR, ^13^C NMR, and HRMS. The preliminary biological assay results indicate that the 4,6-disubstituted pyrimidine and triazine derivatives could maintain high herbicidal activity. It was found that the synthesized compounds could accelerate degradation rates, both in acidic and alkaline soil. Especially, in alkaline soil, the degradation rate of the target compounds accelerated more than 22-fold compared to chlorsulfuron. Moreover, most chlorsulfuron analogs exhibited good crop safety for both wheat and corn at high dosages. This study provided a reference for the further design of new sulfonylurea herbicides with high herbicidal activity, fast degradation rates, and high crop safety.

## 1. Introduction

Chlorsulfuron, one of the classical sulfonylurea herbicides, was developed and marketed in the 1980s and has been widely used as a wheat-field herbicide due to its extremely efficient herbicidal activity and safety for mammals [1].

As acetolactate synthase (ALS) inhibitors, chlorsulfuron blocks the biosynthesis of three essential amino acids (valine, leucine, and isoleucine) and leads to the inhibition of weed growth [2].

China feeds 22% of the world’s population with 7% of the world’s cultivated area [3]. Due to the large demand of food for a growing population, a crop rotation mode has been implemented in some countries, including China [4]. With the large-scale use of sulfonylurea herbicides, some sulfonylurea herbicides show different toxicities of residues on different crop varieties because of their long degradation times and this has caused serious economic losses [5]. Chlorsulfuron, a classical sulfonylurea herbicide that exhibits good safety on wheat but seriously endangers the normal growth of subsequent corn, had its field application suspended by the Ministry of Agriculture of China in 2014 [6].

Sulfonylurea herbicides have a long residual period in alkaline soils. Walker et al. reported that the DT_50_ of chlorsulfuron in sandy soil (at 6% soil moisture) with pH 7.1 was 54.2 days at 10 °C [7]. Thirunarayanan et al. reported that the degradation half-life (DT_50_) of chlorsulfuron was 105.0 days in soil with pH 7.1 at 20 °C and 143.3 days at pH 8.1 [8] Fredrickson et al. reported that the DT_50_ of chlorsulfuron was 10 weeks in soil with pH 7.5 and 12.5 weeks at pH 8.0 [9]. Nicholls et al. reported that the DT_50_ of chlorsulfuron could be 18 months in soils with low microbial activity and high pH [10].

Our group has been working on the problem of the long degradation times of sulfonylurea herbicides. In our previous study, we found that the 5th substituted groups on the benzene ring in chlorsulfuron has a great influence on its degradation rate in acidic soils and concluded for the first time that electron-donating substituents could accelerate degradation rates in acidic soil (Figure 1) [11,12]. Li et al. found that 5-dialkylamino-substituted groups on the benzene ring of chlorsulfuron could greatly accelerate degradation under alkaline conditions [13,14,15]. However, they found that 5-dialkylamino-substituted groups on the benzene ring of chlorsulfuron could endanger the normal growth of wheat and corn through post-emergence treatment [15]. At the same time, they found that heterocyclic amines-based 5-dialkylamino-substituted chlorsulfuron derivatives could improve crop safety of wheat [14], and the crop safety of corn with diethylamino-substituted chlorsulfuron was better than that of dimethylamino-substituted chlorsulfuron through pre-emergence treatment [15].

In recent years, pyrimidine- and triazine-based heterocycles have been reported to exhibit excellent bioactivity in herbicides, especially sulfonylurea herbicides used on wheat and corn [16,17].

In order to obtain sulfonylurea herbicides with good crop safety both for wheat and corn, while maintaining their high herbicidal activities, we synthesized a series of pyrimidine- and triazine-based diethylamino-substituted chlorsulfuron derivatives (Figure 2). As multi-factor (activities, crop safety and soil degradation) relationship should be considered to explore new green sulfonylurea herbicides under different crop rotation modes, we herein systematically evaluated their herbicidal activities, crop safety for both wheat and corn, as well as soil degradation (in both acidic and alkaline soil) with chlorsulfuron as the control.

## 2. Materials and Methods

### 2.1. Instruments and Materials

^1^H and ^13^C NMR were obtained using a 400 MHz Bruker AV 400 spectrometer (Bruker Co., Fallanden, Switzerland). A Rigaku Saturn 70 CCD diffractometer (Rigaku Corporation, Tokyo, Japan) was used to obtain crystal data. Agilent 6520 Q-TOF LC/MS high-resolution mass spectrometer (HRMS) (Agilent Co., Santa Clara, CA, USA) was used to obtain the HRMS.

All reagents for high-performance liquid chromatography (HPLC) were chromatographic grade, reaction reagents were analytical grade and the water phase was double-distilled water. A TU-1810 ultraviolet–visible spectrophotometer (Persee General Analysis Co., Beijing, China) was used to detect the wavelength. HPLC data were obtained on a SHIMADZU LC-20AT (SHIMADZU Co., Kyoto, Japan), equipped with a binary pump (SHIMADZU, LC-20AT), a UV–vis detector (SHIMADZU, SPD-20A), an auto sampler (SHIMADZU, SIL-20A), a Shimadzu shim-pack VP-ODS column (5 μm, 250 × 4.6 mm, C18 reversed phase chromatography) connected to a Shimadzu shim-pack GVP-ODS (10 × 4.6 mm) guard column, a column oven (SHIMADZU, CTO-20AC), and a computer (Dell) to carry out experimental data analysis. Column chromatography purification was carried out using silica gel (200–300 mesh). A SHZ-88 thermostatic oscillator (Jintan Medical Instrument Factory, Changzhou, China), Thermo Scientific Legend Mach 1.6 R centrifuge (Thermo Fisher Scientific Inc., Waltham, MA, USA) and SPX-150B-Z biochemical incubator (Boxun Industrial Co., Shanghai, China) were used in the degradation experiments.

### 2.2. Synthetic Method

The synthetic procedure for compounds **W102**–**W111** is shown in Figure 3.

The synthesis of intermediates 7, 10 and 13 has been reported in previous work [11,18,19,20,21,22,23,24,25].

Intermediate 8 (1.90 mmol) and 9 (2.00 mmol), KI (1.90 mmol), K_2_CO_3_ (3.80 mmol) and 20 mL THF were added to a 100 mL round bottom flask, and magnetically stirred overnight at 0 °C in an ice bath. The progress was monitored by TLC. After completion of the reaction, THF was removed by concentration in a vacuum, and the residue was purified through chromatography on silica gel using petroleum ether/ethyl acetate (*v*/*v* = 6:1) as eluent to obtain intermediate 10.

Intermediate 11 (1.90 mmol), 12 (3.80 mmol), NaH (1.90 mmol) and 20 mL THF were added to a 100 mL round bottom flask, and magnetically stirred overnight at 0 °C in ice bath. Then, 60% NaH (2.06 mmol) was slowly added to the flask. The progress was monitored by TLC. After completion of the reaction, suction filtration was implemented and solid product intermediate 13 was obtained.

Synthesis of target compound **W102**–**W111**.

**W102** was reported in our previous research [11].

Intermediate 10 (1.90 mmol) and 12 (2.00 mmol), DBU (2.85 mmol) and 10 mL of acetonitrile were added to a 25 mL round bottom flask, and magnetically stirred overnight at room temperature. The progress was monitored by TLC. After completion of the reaction, acetone was removed by concentration in a vacuum, and the residue was purified through chromatography on silica gel using petroleum ether/ethyl acetate (*v*/*v* = 5:1) as eluent to obtain the target compounds **W103–W111** (**W105** was not included).

Intermediate 13 (1.90 mmol) and 12 (2.00 mmol), DBU (2.85 mmol) and 10 mL of acetonitrile were added to a 25 mL round bottom flask, and magnetically stirred overnight at room temperature. The progress was monitored by TLC. After completion of the reaction, acetone was removed by concentration in a vacuum. A total of 10 mL of water was added to the residue and the mixture was adjusted to pH 4. The resulting precipitate was obtained by filtration. The product was purified through chromatography on silica gel using petroleum ether/ethyl acetate (*v*/*v* = 5:1) as eluent to obtain the target compounds **W105**.

**W103** White solid product with a yield of 83.2%. Melting point: 179–181 °C. ^1^H NMR (400 MHz, Chloroform-d) *δ* 12.84 (s, 1H, NH), 8.30 (d, *J* = 5.1 Hz, 1H, pyrimidin-H), 7.53 (s, 1H, Ph-H), 7.22 (d, *J* = 8.9 Hz, 1H, Ph-H), 6.72 (d, *J* = 6.5 Hz, 1H, Ph-H), 6.45 (d, *J* = 5.9 Hz, 1H, pyrimidin-H), 4.01 (s, 3H, OCH_3_), 3.40 (q, *J* = 13.9, 6.9 Hz, 4H, CH_2_CH_3_), 1.19 (t, *J* = 7.0 Hz, 6H, CH_2_CH_3_). ^13^C NMR (101 MHz, Chloroform-d) *δ* 170.14, 157.36, 156.43, 148.81, 146.44, 136.21, 132.10, 116.64, 115.75, 115.17, 103.24, 54.72, 44.65, 12.29. HRMS (ESI) calcd for C_16_H_21_ClN_5_O_4_S [M+H]^+^ 414.0997, found 414.0994.

**W104** White solid product with a yield of 81.8%. Melting point: 144–146 °C. ^1^H NMR (400 MHz, Chloroform-d) *δ* 12.98 (s, 1H, NH), 8.46 (d, *J* = 5.1 Hz, 1H, pyrimidin-H), 7.55 (s, 1H, Ph-H), 7.24 (d, *J* = 9.0 Hz, 1H, Ph-H), 6.90 (d, *J* = 5.2 Hz, 1H, Ph-H), 6.76 (d, *J* = 7.1 Hz, 1H, pyrimidin-H), 3.40 (q, *J* = 7.1 Hz, 4H, CH_2_CH_3_), 2.53 (s, 3H, CH_3_), 1.19 (t, *J* = 7.1 Hz, 6H, CH_2_CH_3_). ^13^C NMR (101 MHz, Chloroform-d) *δ* 168.54, 158.30, 156.57, 149.25, 146.38, 136.48, 132.12, 116.59, 115.67, 114.84, 114.84, 44.65, 24.01, 12.27. HRMS (ESI) calcd for C_16_H_21_ClN_5_O_3_S [M+H]^+^ 398.1048, found 398.1049.

**W105** White solid product with a yield of 82.2%. Melting point: 187–189 °C. ^1^H NMR (400 MHz, DMSO-d_6_) *δ* 12.56 (s, 1H, NH), 11.07 (s, 1H, NH), 7.39 (d, *J* = 8.9 Hz, 1H, Ph-H), 7.30 (s, 1H, Ph-H), 6.95 (d, *J* = 9.0 Hz, 1H, Ph-H), 3.98 (s, 6H, OCH_3_), 3.39 (q, *J* = 7.0 Hz, 4H, CH_2_CH_3_), 1.12 (t, *J* = 7.1 Hz, 6H, CH_2_CH_3_). ^13^C NMR (101 MHz, Chloroform-d) *δ* 172.06, 164.52, 147.83, 146.34, 135.85, 132.18, 116.89, 115.75, 115.29, 55.91, 44.75, 12.23. HRMS (ESI) calcd for C_16_H_22_ClN_6_O_5_S [M+H]^+^ 445.1055, found 445.1054.

**W106** White solid product with a yield of 84.1%. Melting point: 197–200 °C. ^1^H NMR (400 MHz, Chloroform-d) *δ* 12.83 (s, 1H, NH), 7.54 (s, 1H, Ph-H), 7.22 (d, *J* = 4.7 Hz, 1H, Ph-H), 6.76 (s, 1H, Ph-H), 5.79 (s, 1H, pyrimidin-H), 3.96 (s, 6H, OCH_3_), 3.40 (q, *J* = 6.8 Hz, 4H, CH_2_CH_3_), 1.19 (t, *J* = 6.9 Hz, 6H, CH_2_CH_3_). ^13^C NMR (101 MHz, Chloroform-d) *δ* 171.48, 155.35, 148.52, 146.26, 136.07, 132.14, 116.76, 115.41, 85.34, 85.28, 54.90, 44.75, 12.23. HRMS (ESI) calcd for C_17_H_23_ClN_5_O_5_S [M+H]^+^ 444.1103, found 444.1102.

**W107** White solid product with a yield of 82.6%. Melting point: 226–228 °C. ^1^H NMR (400 MHz, DMSO-d_6_) *δ* 13.35 (s, 1H, NH), 10.81 (s, 1H, NH), 8.75 (d, *J* = 5.4 Hz, 1H, pyrimidin-H), 8.11 (d, *J* = 8.2 Hz, 2H, Ph-H), 7.79 (d, *J* = 5.4 Hz, 1H, pyrimidin-H), 7.39 (d, *J* = 8.1 Hz, 2H, Ph-H), 7.35 (d, *J* = 8.9 Hz, 1H, Ph-H), 7.32 (d, *J* = 3.1 Hz, 1H, Ph-H), 6.93 (dd, *J* = 9.0, 3.1 Hz, 1H, Ph-H), 3.39 (dd, *J* = 13.3, 6.3 Hz, 4H, CH_2_CH_3_), 2.41 (s, 3H, CH_3_), 1.12 (t, *J* = 7.0 Hz, 6H, CH_2_CH_3_). ^13^C NMR (101 MHz, DMSO-d_6_) *δ* 169.58, 164.18, 163.98, 159.27, 146.43, 141.73, 140.73, 134.67, 132.37, 129.74, 127.08, 115.67, 114.90, 111.50, 105.93, 44.37, 21.40, 12.57. HRMS (ESI) calcd for C_22_H_25_ClN_5_O_3_S [M+H]^+^ 474.1361, found 474.1360.

**W108** White solid product with a yield of 84.8%. Melting point: 231–233 °C. ^1^H NMR (400 MHz, DMSO-d_6_) *δ* 12.87 (s, 1H, NH), 10.78 (s, 1H, NH), 8.70 (d, *J* = 4.9 Hz, 1H, pyrimidin-H), 8.16 (s, 1H, Thio-*H*), 7.98 (d, *J* = 3.7 Hz, 1H, Thio-*H*), 7.72 (d, *J* = 4.8 Hz, 1H, Thio-*H*), 7.36 (d, *J* = 8.9 Hz, 1H, Ph-H), 7.32 (s, 1H, Ph-H), 6.94 (d, *J* = 8.3 Hz, 1H, Ph-H), 5.77 (s, 1H, pyrimidin-H), 3.43–3.36 (m, 4H, CH_2_CH_3_), 1.12 (t, *J* = 6.3 Hz, 6H, CH_2_CH_3_). HRMS (ESI) calcd for C_19_H_21_ClN_5_O_3_S_2_ [M+H]^+^ 465.0769, found 466.0768.

**W109** White solid product with a yield of 80.9%. Melting point: 185–187 °C. ^1^H NMR (400 MHz, Chloroform-d) *δ* 13.34 (s, 1H, NH), 7.57 (s, 1H, Ph-H), 7.28 (d, *J* = 3.7 Hz, 1H, Ph-H), 6.77 (d, *J* = 5.8 Hz, 1H, Ph-H), 6.31 (s, 1H, pyrimidin-H), 3.97 (s, 3H, OCH_3_), 3.42 (dd, *J* = 7.0 Hz, 4H, CH_2_CH_3_), 2.44 (s, 3H, CH_3_), 1.21 (t, *J* = 7.1 Hz, 6H, CH_2_CH_3_). ^13^C NMR (101 MHz, Chloroform-d) *δ* 170.69, 167.79, 155.90, 148.83, 146.29, 136.36, 132.09, 116.62, 115.89, 115.21, 101.80, 54.48, 44.73, 23.46, 12.25. HRMS (ESI) calcd for C_17_H_23_ClN_5_O_4_S [M+H]^+^ 428.1154, found 428.1153.

**W110** White solid product with a yield of 83.5%. Melting point: 199–201 °C. ^1^H NMR (400 MHz, CDCl_3_) *δ* 13.32 (s, 1H, NH), 7.77 (s, 1H, NH), 7.54 (d, *J* = 2.6 Hz, 1H, Ph-H), 7.23 (d, *J* = 8.8 Hz, 1H, Ph-H), 6.78 (s, 1H, Ph-H), 6.73 (dd, *J* = 8.8, 2.7 Hz, 1H, pyrimidin-H), 3.41 (q, *J* = 6.9 Hz, 4H, CH_2_CH_3_), 2.48 (s, 6H, CH_3_), 1.20 (t, *J* = 7.0 Hz, 6H, CH_2_CH_3_). ^13^C NMR (101 MHz, Chloroform-d) *δ* 168.49, 156.31, 149.08, 146.34, 136.43, 132.05, 116.49, 115.61, 115.25, 114.92, 44.61, 23.67, 12.27. HRMS (ESI) calcd for C_17_H_23_ClN_5_O_3_S [M+H]^+^ 412.1205, found 412.1203.

**W111** White solid product with a yield of 85.1%. Melting point: 209–210 °C. ^1^H NMR (400 MHz, CDCl_3_) *δ* 12.29 (s, 1H, NH), 7.54 (s, 1H, Ph-H), 7.31 (s, 1H, Ph-H), 6.78 (s, 1H, Ph-H), 6.50 (s, 1H, pyrimidin-H), 4.05 (s, 3H, CH_3_), 3.40 (dd, *J* = 13.7, 6.8 Hz, 4H, CH_2_CH_3_), 1.19 (t, *J* = 6.9 Hz, 6H, CH_2_CH_3_). ^13^C NMR (101 MHz, DMSO-d_6_) *δ* 171.04, 160.35, 156.70, 148.24, 146.45, 136.33, 132.74, 117.59, 114.69, 114.49, 101.82, 55.68, 44.53, 12.46. HRMS (ESI) calcd for C_16_H_20_Cl_2_N_5_O_4_S [M+H]^+^ 448.0608, found 448.0609.

### 2.3. X-ray Diffraction

The crystal of **W110** was obtained by self-evaporation of the mixture solvent of dichloromethane and methyl tert-butyl ether (*v*/*v* = 1:2). It was colorless and analyzed using X-ray diffraction. Crystal data of compound **W110** can be found in the Appendix A.

### 2.4. Herbicidal Activity

The herbicidal activity test methods of the target compounds referred to the literature [26,27,28]. Pot trials for herbicidal activities against *Echinochloa crusgalli*, *Digitaria sanguinalis*, *Amaranthus tricolor*, and *Brassica campestris* were tested with chlorsulfuron as a positive control.

Pre-emergence treatment: An appropriate amount of culture soil (vermiculite:loam:fertilizer soil = 1:1:1) was added to a paper cup with a height of 7.5 cm. The chemicals were sprayed at the dosage of 15 g·ha^−1^ and 150 g·ha^−1^, respectively. Weed seeds were planted and water was added to adjust soil moisture. Samples were cultivated in a greenhouse at 25 ± 2 °C. Plants were watered regularly to ensure normal growth of weeds. The fresh weight of the above-ground weeds was measured after 28 days, and the inhibition rate was calculated and the herbicidal activity of the target compound was indicated by the fresh weight inhibition percentage. The variance analysis of the data was performed using the Duncan multiple comparisons with DPS16.5 software.

Post-emergence treatment: Seeding was conducted following adjustment of the moisture and temperature, and the tested chemicals were sprayed until the weeds grew up to the one leaf and one stem stage. The fresh weight of the above-ground weeds was measured after 28 days, and the inhibition rate was calculated and the herbicidal activity of the target compound was indicated by the fresh weight inhibition percentage. The variance analysis of the data was performed using the Duncan multiple comparisons with DPS16.5 software.

### 2.5. Soil Degradation

The culture method was consistent with our previous reports [11,12,13,14,15,23]. The detailed experimental procedure can be found in the Appendix A. Soil degradation steps are described here briefly.

Soil was selected from Jiangxi (pH 5.46) and Hebei (pH 8.39) provinces to develop degradation studies [29]. The properties of the tested soils are listed in Table 1. HPLC analysis conditions were measured using acetonitrile, ultrapure water (pH 3.0) and chromatographically pure methanol as the mobile phase. The retention time remained between 10–20 min. Standard curves were established with an injection volume of 10 μL at 20 °C and concentration range was between 200 ug mL^−1^ and 0.025 ug mL^−1^. Additionally, the concentrations of the test compounds in the 100 mL conical flask were 0.5, 2, and 5 mg kg^−1^ (in acetonitrile) for 20 g of soil. Each concentration of sample was repeated 5 times. In addition, the recovery rates remained between 70 and 110% with a coefficient of variation (RSD) no more than 5%. And for cultivation of samples, the concentration of each sample was 5 mg kg^−1^ in soil. Water was added to adjust soil moisture and the samples were cultivated at 25 ± 1 °C by a biochemical incubator at 80% humidity under dark conditions. Soil samples of three replicates were collected at six different times. Finally, according to the kinetic equation Ct = C_0_ × e^−kt^, DT_50_ values were calculated following the formula: DT_50_ = ln(2)/k (the analytical data for verification of recovery rates in various concentrations can be found in Appendix A).

### 2.6. Crop Safety Assay

Chlorsulfuron is a classic sulfonylurea herbicide used in wheat fields. Its residual in soil seriously affects the normal growth of following crop seedlings. Corn is one of the crops planted after wheat. In this study, wheat (Xinong 529) and corn (Xindan 66) were used as test crops, and chlorsulfuron was used as control. The culture method was consistent with previous reports [15,23,27].

Methods of plant cultivation: An appropriate amount of culture soil (vermiculite: loam: fertilizer soil = 1:1:1) was added to a paper cup with a height of 7.5 cm. Seeds were planted and water was added to adjust the soil moisture. Samples were covered with plastic wrap and cultivated in a greenhouse at 25 ± 2 °C.

Pre-emergence treatment: The chemicals were sprayed at concentrations of 15 g·ha^−1^ and 150 g·ha^−1^. Seeds were planted and water was added to adjust the soil moisture. Samples were cultivated in a greenhouse at 25 ± 2 °C. Plants were watered regularly to ensure the normal growth of weeds. The fresh weight of the cover crops was measured after several days (22 days for wheat and 16 days for corn).

Post-emergence treatment: The chemicals were sprayed when the wheat grew to the 4-leaf stage. For corn, the safety assay began when the corn grew to the 3-leaf stage. The fresh weight of the cover crops was measured after several days (28 days for wheat and 23 days for corn).

The fresh weight of the above-ground crops was measured after several days, and the inhibition rates of fresh weight were used to represent the safety of the crops. The variance analysis of the data was performed using the Duncan multiple comparisons with SPSS 22.0.

## 3. Results and Discussion

### 3.1. Chemistry

Compounds **W103**–**W111** were characterized by melting points, ^1^H NMR, ^13^C NMR and HRMS (all the spectra are listed in Appendix A).

For pyrimidine- and triazine-substituted chlorsulfuron derivatives, the ^1^H NMR spectra of compound **W103** and **W105** was taken as an example.

^1^H NMR spectra of compound **W103**: The chemical shifts of NH in the sulfonylurea bridge was at *δ* 12.84. The chemical shift of H on pyrimidine ring was displayed at *δ* 8.30 and *δ* 6.45, while 2-CH_3_O was at *δ* 4.01. The aromatic hydrogens presented at δ 7.53, *δ* 7.22 and *δ* 6.72, respectively. The chemical shifts of CH_3_ and CH_2_ on 5-amino were at *δ* 1.19 and *δ* 3.40, respectively.

^1^H NMR spectra of compound **W105**: The chemical shifts of NH in the sulfonylurea bridge was at *δ* 12.56 and *δ* 11.07. The chemical shift of H was displayed at *δ* 3.98 (CH_3_O) on the triazine ring. The aromatic hydrogens presented at δ 7.39, *δ* 7.30 and *δ* 6.95, respectively. The chemical shifts of CH_3_ and CH_2_ on 5-amino were at *δ* 1.12 and *δ* 3.39, respectively.

### 3.2. X-ray Diffraction

A suitable crystal was selected and obtained using a Rigaku Saturn 70 CCD diffractometer. The crystal was kept at 113.15 K during data collection. Using Olex2 [30], the structure was solved with the ShelXT [31] structure solution program using Intrinsic Phasing and refined with the ShelXL [32] refinement package using least squares minimization.

The crystal structure of **W110** is shown in Figure 4 (CCDC no. 2142702).

The sum of the angles O(3)−C(11)−N(2) [123.7(7)°], O(3)−C(11)−N(3) [121.8(7)°], and N(2)−C(11)−N(3)[114.5(6)°] was 360°, indicating the plane sp^2^ hybridization state of the C(11) atom.

The bonds angles O(1)−S(1)−N(2), O(2)−S(1)−C(9), and N(2)−S(1)−C(9) were 110.5(3)°, 109.8(3)°, and 104.0(3)°, respectively, which indicated that the state of S(1) atom was sp^3^ hybridization. The bond lengths of N(1)–C(4) and N(1)–C(2) were 1.468(10) nm and 1.462(10) nm, respectively. The bond length of N(1)–C(5) was 1.370(10) nm, which was shorter than the general C–N bonds. It was speculated that the shorter bond was caused by the transfer of π electrons to the benzene ring. The dihedral angle of the pyrimidine ring and benzene ring was 87.938(225), which were non-planar.

### 3.3. Herbicidal Activity

The herbicidal activities of title compounds against *Echinochloa crusgalli*, *Digitaria sanguinalis*, *Amaranthus tricolor*, and *Brassica campestris* were screened through pre- and post-emergence treatment at 15 g·ha^−1^ and 150 g·ha^−1^ with chlorsulfuron as a positive control.

As shown in Table 2, the herbicidal activities of **W103**, **W104**, **W107** and **W108** are not ideal. For pre-emergence treatment, at concentration of 150 g·ha^−1^, the inhibition rates of **W104**, **W103**, **W107** and **W108** against *Brassica campestris* were 70.8%, 13.9%, 11.8% and 2.4%, respectively. The inhibition rates against *Echinochloacrusgalli* were 65.3%, 40.1%, 0% and 0%. It seemed that mono substituents on the 4-position of the pyrimidine ring derivatives may have decreased the herbicidal activity. The larger the volume of the mono substituent, the lower the herbicidal activity. Overall scanning of the herbicidal activities of **W106**, **W109**, **W110** and **W111** showed that they maintained high herbicidal activities compared with chlorsulfuron. At a concentration of 150 g·ha^−1^, the inhibition rates against *Brassica campestris* were 100%, 100%, 94.2% and 100%, respectively, which were comparable to chlorsulfuron (100%) under post-emergence treatments. At a concentration of 150 g·ha^−1^, the inhibition rates of **W106** and **W109** against *Amaranthus tricolor* surpassed chlorsulfuron under both pre- and post-emergence treatments. For post-emergence treatments, the inhibition rates of **W109**, **W110** and **W111** against *Echinochloa crusgalli* were 98.4%, 79.4% and 98.4%, respectively, which were higher than chlorsulfuron (60.2%). For compound **W105**, the inhibition rates against *Brassica campestri* and *Amaranthus tricolor* were 88.1% and 91.2%, respectively, at 150 g·ha^−1^ through pre-emergence treatments. The inhibition rate of **W105** against *Echinochloa crusgalli* was 90.6%, which was better than chlorsulfuron (60.2%).

Based on the data above, we found that the 4,6-disubstituted pyrimidine and triazine derivatives could maintain a high herbicidal activity.

### 3.4. Soil Degradation

Anderson et al. reported that chlorsulfuron had long persistance in soil with pH 5.7 and 6.1 [33]. Thirunarayanan et al. reported that the degradation half-life (DT_50_) of chlorsulfuron was 38.1 days in soil with pH 6.2 at 20 °C [8]. Fredrickson et al. reported that the DT_50_ of chlorsulfuron in soil with pH 5.6 was 1.9 weeks and, in soil with pH 6.3, it was 2.7 weeks at 25 °C [8]. Li found that the DT_50_ of chlorsulfuron was 12.91 days in soil with pH 5.40 at 25 °C, and the DT_50_ of diethylamino-substituted chlorsulfuron was 1.60 days [11].

With chlorsulfuron as the positive control, the degradation of compounds (W) were firstly investigated in soil at pH 5.46. The kinetic parameters for acidic soil degradation are listed in Table 3.

As shown in Table 3, the DT_50_ of **W102**–**W111** were 1.87–15.13 days, while the DT_50_ of chlorsulfuron was 17.64 days in acidic soil (pH 5.46). The degradation rates of the W-series compounds were 1.17–9.43 times faster than that of chlorsulfuron.

The DT_50_ of **W102** was 1.87 days, which was 9.43 times faster than that of chlorsulfuron (17.64 days). For the other compounds, the DT_50_ of **W103** was 1.89 days (9.33 times faster), the DT_50_ of **W104** was 2.87 days (6.15 times faster), the DT_50_ of **W105** was 3.22 days (5.48 times faster), the DT_50_ of **W106** was 11.13 days (1.58 times faster), the DT_50_ of **W107** was 15.13 days (1.17 times faster), the DT_50_ of **W109** was 4.03 days (4.38 times faster), the DT_50_ of **W110** was 6.60 days (2.67 times faster), and the DT_50_ of **W111** was 10.15 days (1.74 times faster).

Comparing the degradation rates with the diethylamino-substituted chlorsulfuron derivative (**W102**), it was found that the degradation rates can be adjusted by introducing pyrimidine and triazine heterocycles into acidic soil.

Walker et al. reported that the DT_50_ of chlorsulfuron in sandy soil (at 6% soil moisture) with pH 7.1 at 20 °C was 56.0 days [7]. Thirunarayanan et al. reported that the DT_50_ of chlorsulfuron in soil with pH 7.7 was 136.6 days at 20 °C and 231.7 days at 10 °C [8] Fredrickson et al. reported that he DT_50_ of chlorsulfuron was 12.5 weeks at pH 8.0 in sandy loam [9]. Nicholls et al. reported that the DT_50_ of chlorsulfuron could be 18 months in soil with low microbial activity and a high pH [10]. Li reported that the DT_50_ of chlorsulfuron was 158 days at pH 8.39, and DT_50_ of diethylamino-substituted chlorsulfuron was 6.39 days [15].

Considering the slow degradation of chlorsulfuron in alkaline soil, the degradation of compounds (W) were investigated in soil with pH 8.39. The kinetic parameters for alkaline soil degradation are listed in Table 4.

As shown in Table 4, the DT_50_ of **W102**–**W111** was 5.81–6.97 days while the DT_50_ of chlorsulfuron was 157.53 days in alkaline soil (pH 8.39). The degradation rates of the W-series compounds were 22.6–27.1 times faster than that of chlorsulfuron.

As the alkaline soil degradation results showed, the degradation rates of **W102**–**W111** were reduced to 5.81–6.97 days, which were accelerated by 22.6–27.1-fold as compared to chlorsulfuron (157.53 days). The DT_50_ of **W102** was 6.35 days, which was 24.81 times faster than that of chlorsulfuron. For the other compounds, the DT_50_ of **W103**, **W104**, **W106** and **W111** were 6.30, 6.33, 6.58 and 6.97 days, respectively. The DT_50_ of **W105** was 5.99 days (26.3 times faster), the DT_50_ of **W109** was 5.97 days (26.4 times faster), and the DT_50_ of **W110** was 5.81 days (27.1 times faster). The degradation rates of the synthesized compounds were accelerated dramatically compared with chlorsulfuron.

Considering the degradation rates of compounds (W) in soil with pH 5.46 and pH 8.39, we could find that introduction of the pyrimidine and triazine heterocycles into the 5-diethylamino substituted chlorsulfuron could accelerate the degradation rate in soil. Moreover, the heterocyclic amine is not the key factor affecting its degradation in alkaline soil.

### 3.5. Crop Safety

The crop safety of the target compounds is shown in Table 5 (for wheat) and Table 6 (for corn).

As the data show, crop safety was evaluated at 30 and 60 g·ha^−1^ with chlorsulfuron as the control. The inhibition rates, presented in Table 5 and Table 6, indicate that most compounds were safe for wheat and corn.

Chlorsulfuron is a popular sulfonylurea herbicide applied to wheat fields, but it causes a certain degree of damage to corn [7,34]. In 2018, Li found that diethylamino-substituted chlorsulfuron could cause a certain degree of damage to wheat (Jima 22) [14]. Li also found that diethylamino-substituted chlorsulfuron derivative was less safe for wheat (Xinong 529) and corn (Xindan 66) through post-emergence treatments [15].

In this research, a series of pyrimidine- and triazine-based diethylamino-substituted chlorsulfuron derivatives were tested using a crop safety assay for wheat (Xinong 529) and corn (Xindan 66).

The inhibition rates of compounds (W) indicated that most compounds were safe for wheat. At 60 g·ha^−1^, the inhibition rate of **W102** was 23.3% under pre-emergence treatment. Compounds **W103**–**W111** improved the safety of diethylamino-substituted chlorsulfuron derivatives for pre-emergence treatment on wheat. It was found that the inhibition rates of **W103**, **W105**, **W106** and **W107** were 0% under pre and post emergence treatments. They indeed exhibited good crop safety compared with chlorsulfuron (6.3% and 0%). For other compounds, the inhibition rates of **W104** and **W111** were 1.2% and 0%, 0% and 12.1%, respectively.

In the case of corn, chlorsulfuron exhibited a certain degree of damage to the corn. At 60 g·ha^−1^, the inhibition rates of chlorsulfuron was 46.7% under pre-emergence treatment. The inhibition rates of **W102**, **W109** and **W110** were 36.6%, 59.7% and 47.5%, respectively; they exhibited poor safety for corn. While the inhibition rates of **W107** and **W111** decreased to 1.8% and 19.2%, respectively. Moreover, for **W103**, **W104**, **W105** and **W106**, the inhibition rates were all 0%, which surpassed chlorsulfuron (46.7%).

It was found that such pyrimidine- and triazine-derivated sulfonylureas could exhibit good crop safety on wheat and improve crop safety on corn.

Based on the above results, we believe that compounds such as **W105** and **W106** are potential green sulfonylurea herbicides for wheat and corn.

## 4. Conclusions

In order to explore sulfonylurea herbicides with high herbicidal activities, controllable degradation, and good crop safety, a series of chlorsulfuron derivatives were designed and synthesized by introducing pyrimidine and triazine heterocycles.

Pot trials for herbicidal activities indicated that the 4,6-disubstituted pyrimidine and triazine derivatives could maintain high herbicidal activity, while 4-monosubstituted derivatives exhibited poor herbicidal activities. It was found that pyrimidine- and triazine-substituted chlorsulfuron derivatives could accelerate degradation rates, both in acidic and alkaline soil. Especially, in alkaline soil, the degradation rate of the target compounds accelerated more than 22-fold compared to chlorsulfuron. Additionally, heterocyclic amine is not the key factor affecting its degradation in alkaline soil. Moreover, the W-series compounds exhibited good crop safety for wheat and could improve crop safety for corn. In combination with structures, herbicidal activities, crop safety, as well as soil degradation (in both acidic and alkaline soil), it could be concluded that compounds such as **W105** and **W106** could be potentially employed as sulfonylurea herbicides on wheat and corn. Compounds such as **W111** are potential green sulfonylurea herbicides for pre-emergence treatment for wheat and corn. Further research will be continued to discover new sulfonylurea herbicides to meet the needs of crop rotation modes.

## Figures and Tables

**Figure 1 molecules-27-02362-f001:**
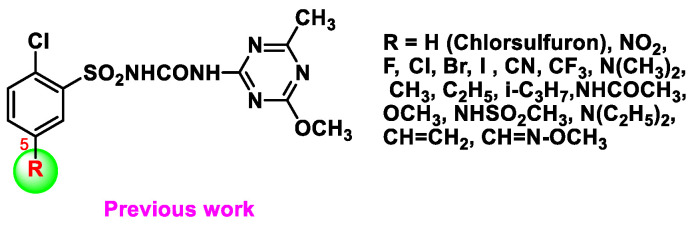
The 5th substituted groups on the benzene ring in chlorsulfuron.

**Figure 2 molecules-27-02362-f002:**
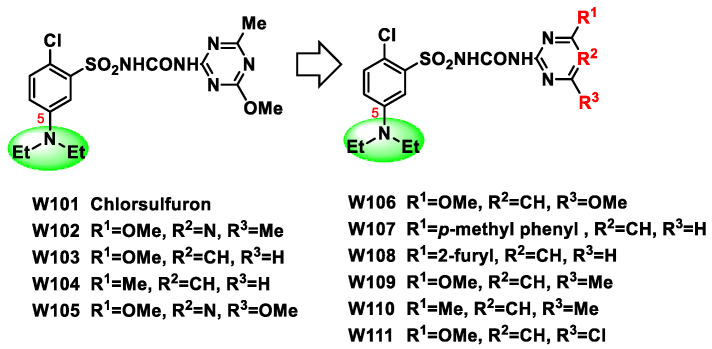
Design of the title compounds W.

**Figure 3 molecules-27-02362-f003:**
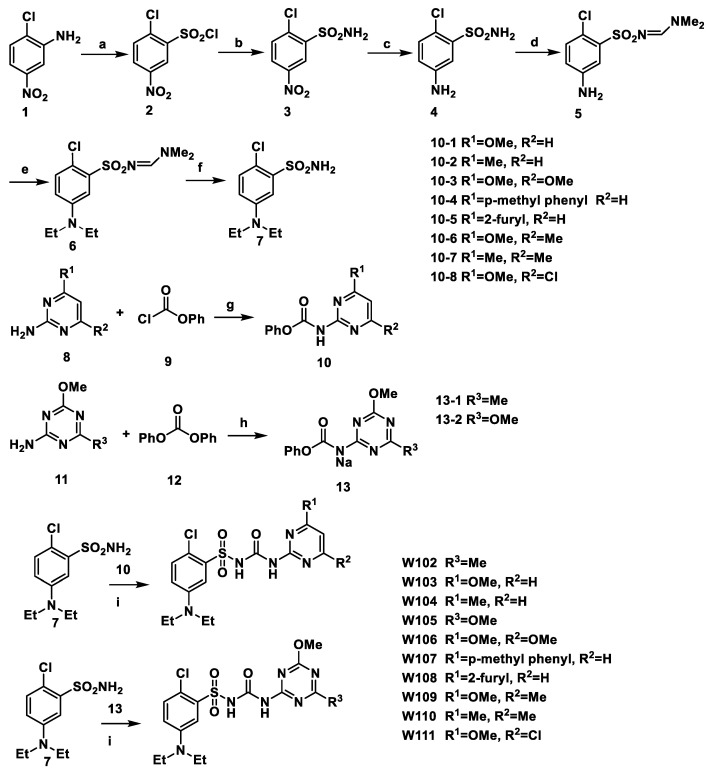
The synthesis of compounds **W102**–**W111**. Reagents and conditions: (a) H_2_O, HCl, NaNO_2_→H_2_O, HCl, CuCl_2_, NaHSO_3_, −5 °C; (b) 28% NH_3_ H_2_O, THF, 0 °C→RT (room temperature), overnight; (c) Fe, HCl, C_2_H_5_OH, H_2_O, reflux; (d) DMF-DMA (*N*,*N*-dimethylformamide dimethyl acetal), CH_2_Cl_2_; (e) IC_2_H_5_, K_2_CO_3_, DMF, 65 °C; (f) 80% H_2_NNH_2_•H_2_O, C_2_H_5_OH; (g) KI, K_2_CO_3_, THF, 0 °C→RT; (h) 60% NaH, THF, 0 °C→RT; (i) DBU (1,8-diazabicyclo[5.4.0]undec-7-ene), CH_3_CN.

**Figure 4 molecules-27-02362-f004:**
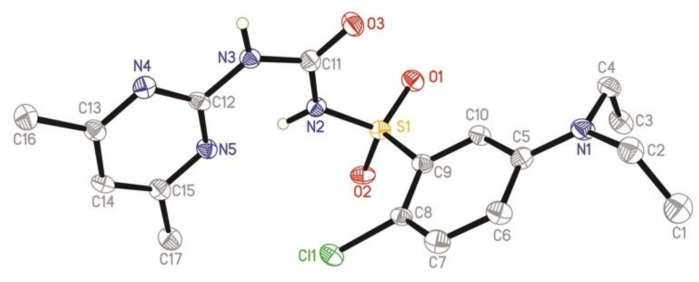
Crystal structure **W110**.

**Table 1 molecules-27-02362-t001:** Analytical data of soils.

Soils	Soil Texture	pH	Cation Exchange Capacity (cmol^+^·kg^−1^)	Organic Matter (g·kg^−1^)	Soil Separation (mm)/Mechanical Composition (%)								
Alkaline soils	Loam	8.39	7.30	19.4	1–2	0.5–1	0.025–0.5	0.05–0.02	0.02–0.002	<0.002	0.25–0.05	2.0–0.05	0.05–0.002
					0.795	2.46	2.33	7.90	28.6	28.2	29.7	35.3	36.5
Acidic soils	Sandy loam	5.46	14.4	8.37	0.075	0.381	0.708	12.5	17.9	10.5	57.9	59.1	30.4

**Table 2 molecules-27-02362-t002:** Herbicidal activities of the title compounds (W) at a concentration of 15 g·ha^−1^ and 150 g·ha^−1.^

Compd.	Concentration (g·ha ^−1^)	Herbicidal Activity, Inhibition Rates (%)							
		Pre-Emergence Treatment				Post-Emergence Treatment			
		*Brassica* *campestris*	*Amaranthus* *tricolor*	*Echinochloa* *crusgalli*	*Digitaria* *sanguinalis*	*Brassica* *campestris*	*Amaranthus* *tricolor*	*Echinochloa* *crusgalli*	*Digitaria* *sanguinalis*
**W101** (Chlorsulfuron)	15	93.7	91.2	57.5	51.0	87.9	53.3	39.1	40.5
	150	96.9	94.1	92.1	61.3	100.0	80.8	60.2	79.7
**W102**	15	82.5	70.6	89.0	47.6	88.4	14.2	38.3	20.0
	150	98.7	95.9	96.8	65.1	100	98.3	59.4	82.4
**W103**	15	0	0	8.2	0	0.0	0	0	0
	150	13.9	52.9	40.1	20.9	37.2	28.3	28.9	0
**W104**	15	4.9	41.2	29.5	21.1	40.9	11.7	0	16.2
	150	70.8	70.6	65.3	42.0	97.5	61.7	33.9	48.6
**W105**	15	81.9	64.7	46.8	28.4	87.7	41.7	27.3	0
	150	88.1	91.2	90.9	52.1	100.0	73.3	90.6	86.5
**W106**	15	57.2	85.3	73.2	0	78.3	53.3	31.3	0
	150	87.5	97.1	87.4	47.8	100	90.8	76.6	79.7
**W107**	15	0	29.4	0	0	0.0	0	0	0
	150	11.8	64.7	0	23.5	15.8	3.3	11.6	6.7
**W108**	15	0	0	0	0	0	0	0	0
	150	2.4	0.0	0	14.2	16.6	9.2	10.8	0
**W109**	15	74.7	83.3	35.0	18.3	79.4	56.7	86.3	18.1
	150	96.3	98.3	76.1	57.4	100	100	98.4	60.1
**W110**	15	65.7	58.3	20.8	0	35.9	16.3	66.3	0
	150	88.4	88.3	80.5	52.5	94.2	86.5	79.4	42.6
**W111**	15	55.7	66.7	3.2	0	52.6	30.8	52.0	0
	150	92.6	83.3	71.0	7.0	100	66.3	98.4	61.5

**Table 3 molecules-27-02362-t003:** Kinetic parameters for acidic soil (pH 5.46) degradation.

Compound	Kinetic Equations of Soil Degradation	Correlation Coefficient (R^2^)	DT_50_ (Days)
**W101** (Chlorsulfuron)	*C_t_* = 3.8317e^−0.0393t^	0.9477	17.64
**W102**	*C_t_* = 3.6838e^−0.3700t^	0.9866	1.87
**W103**	*C_t_* = 3.4989e^−0.3666t^	0.9664	1.89
**W104**	*C_t_* = 3.2440e^−0.2414t^	0.9629	2.87
**W105**	*C_t_* =3.2659e^−0.2150t^	0.9575	3.22
**W106**	*C_t_* = 3.3608e^−0.0623t^	0.9973	11.13
**W107**	*C_t_* = 3.5961e^−0.0458t^	0.9909	15.13
**W109**	*C_t_* = 3.6979e^−0.1721t^	0.9760	4.03
**W110**	*C_t_**=* 3.4616e^−0.1051t^	0.9594	6.60
**W111**	*C_t_**=* 3.4271e^−0.0683t^	0.9614	10.15

**Table 4 molecules-27-02362-t004:** Kinetic parameters for alkaline soil (pH 8.39) degradation.

Compound	Kinetic Equations of Soil Degradation	Correlation Coefficient (R^2^)	DT_50_ (Days)
**W101** (Chlorsulfuron)	*C_t_* = 4.3043e^−0.0044t^	0.9899	157.53
**W102**	*C_t_* = 5.1771e^−0.1091t^	0.9888	6.35
**W103**	*C_t_* = 4.9927e^−0.1100t^	0.9839	6.30
**W104**	*C_t_* = 4.6088e^−0.1095t^	0.9765	6.33
**W105**	*C_t_* =5.3757e^−0.1158t^	0.9795	5.99
**W106**	*C_t_* = 5.0349e^−0.1054t^	0.9787	6.58
**W109**	*C_t_* = 5.4294e^−0.1162t^	0.9761	5.97
**W110**	*C_t_**=* 5.5261e^−0.1193t^	0.9857	5.81
**W111**	*C_t_**=* 4.3883e^−0.0995t^	0.9892	6.97

**Table 5 molecules-27-02362-t005:** Crop safety of target compounds on wheat.

Compound	Concentration (g·ha^−1^)	Wheat (Xinong 529)								
		Pre. (22 Days after Treatment)					Post. (28 Days after Treatment)			
		Fresh Weight g/10 Strains	Analysis of Variance ^a^		Inhibition (%)	Fresh Weight g/10 Strains		Analysis of Variance ^a^		Inhibition (%)
			5%	1%				5%	1%	
	0	2.879	ab	AB	-	3.168		c	B	-
**W101** (Chlorsulfuron)	30	3.043	a	A	0	3.287		a	A	0
	60	2.697	abcd	ABC	6.3	3.287		ab	AB	0
**W102**	30	2.340	de	CD	18.7	3.182		ab	AB	0
	60	2.207	e	D	23.3	3.236		ab	AB	0
**W103**	30	2.942	ab	AB	0	3.427		ab	AB	0
	60	3.070	a	A	0	3.843		abc	AB	0
**W104**	30	2.927	ab	AB	0	3.530		abc	AB	0
	60	2.843	abc	AB	1.2	3.443		abc	AB	0
**W105**	30	2.924	ab	AB	0	3.447		abc	AB	0
	60	2.931	ab	AB	0	3.423		abc	AB	0
**W106**	30	2.969	ab	AB	0	3.287		abc	AB	0
	60	2.917	ab	AB	0	3.575		abc	AB	0
**W107**	30	2.950	ab	AB	0	3.244		abc	AB	0
	60	3.080	a	A	0	3.283		abc	AB	0
**W109**	30	2.747	a	AB	4.6	2.703		bcd	BC	14.7
	60	2.544	ab	AB	11.6	2.674		cd	BC	15.6
**W110**	30	2.607	ab	AB	9.4	3.082		abc	AB	2.7
	60	2.557	ab	AB	11.2	2.996		abc	AB	5.4
**W111**	30	2.915	a	A	0	3.105		abc	AB	2.0
	60	2.890	a	A	0	2.784		bc	BC	12.1

^a^ Among the averages, the same letter indicates that there was no significant difference, and different letters indicate that there was a significant difference.

**Table 6 molecules-27-02362-t006:** Crop safety of target compounds on corn.

Compound	Concentration (g·ha^−1^)	Corn (Xindan 66)								
		Pre. (16 Days after Treatment)					Post. (23 Days after Treatment)			
		Fresh Weight g/5 Strains	Analysis of Variance ^a^		Inhibition (%)	Fresh Weight g/5 Strains		Analysis of Variance ^a^		Inhibition (%)
			5%	1%				5%	1%	
	0	6.312	abc	ABCD	-	8.073		a	A	-
**W101** (Chlorsulfuron)	30	4.043	cdef	BCDE	35.9	7.932		a	A	1.7
	60	3.361	ef	DE	46.7	7.370		a	A	8.7
**W102**	30	4.060	cdef	BCDE	35.7	7.146		a	A	11.5
	60	4.000	cdef	BCDE	36.6	7.109		a	A	11.9
**W103**	30	7.080	ab	ABC	0	8.419		a	A	0
	60	6.716	ab	ABCD	0	8.433		a	A	0
**W104**	30	7.622	a	A	0	8.493		a	A	0
	60	6.787	ab	ABC	0	8.644		a	A	0
**W105**	30	6.437	abc	ABCD	0	8.329		a	A	0
	60	7.222	ab	AB	0	7.764		a	A	3.8
**W106**	30	7.109	ab	ABC	0	8.138		a	A	0
	60	6.360	abc	ABCD	0	7.454		a	A	7.7
**W107**	30	6.902	ab	ABC	0	8.389		a	A	0
	60	6.200	abcd	ABCD	1.8	8.377		a	A	0
**W109**	30	3.524	cd	BCD	44.2	7.871		a	A	2.5
	60	2.547	de	DE	59.7	7.514		a	A	6.9
**W110**	30	3.521	cd	BCD	44.2	7.132		a	A	11.7
	60	3.314	cde	BCD	47.5	7.013		a	A	13.1
**W111**	30	5.671	ab	A	10.2	8.095		a	A	0
	60	5.102	ab	AB	19.2	8.077		a	A	0

^a^ Among the averages, the same letter indicates that there was no significant difference, and different letters indicate that there was a significant difference.

## Data Availability

Data are contained within the article and the Appendix A.

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
