# Peer review of "Synthesis, Herbicidal Activity, Crop Safety and Soil Degradation of Pyrimidine- and Triazine-Substituted Chlorsulfuron Derivatives"

_molecules, 2022, doi:10.3390/molecules27072362_

Round 1

Reviewer 1 Report

The presented manuscript reports interesting data on the herbicidal activity, crop safety and soil degradation of pyrimidine- and triazine-substituted derivatives of chlorsulfuron. At the same time, there is no information on the synthesis (except the general scheme) and characterization of these derivatives. In addition to the fact that the synthesis of these compounds is stated in the title of the article, it is certainly of interest to a wider readership compared to data on their degradation in specific soils, crop safety, and even herbicidal activity, which depend on various factors and, as a rule, are less reproducible compared to synthesis. The available literature describes the synthesis of only one compound with the triazine ring W102 (Ref. 9 and 15) and the synthesis of compounds 7 (Ref. 9) and 13 (Ref. 15) used to obtain another compound with a triazine ring W105, while any data on the synthesis of compounds with a pyrimidine ring (except for one compounds of series 10 – Ref. 16 + F. Meng, et al., J. Mol. Struct., 2022, 1260, 132756) are absent. Ref. 17 provided by the authors is not available to the vast majority of readers. Therefore, the synthesis and characterization of all new compounds should be given.

At the bottom of Fig. 2, the designations of the substituents in the heterocyclic ring change from R1, R2, and R3 in the starting compounds to R4, R5, and R6, although the substituents themselves do not change. This is confusing and should be fixed.

Do the authors have data on the products of degradation of the obtained compounds in the soil and their possible further effect on crop safety?

Some minor problems:

Page 1, 3-rd paragraph – “Our country” – it may not be clear to the reader. “China” is better and more correct.

Page 2, 2-nd paragraph – “the 5th substituted groups” – there is no numeration of atoms in Fig. 1.

Reviewer 2 Report

The manuscript "Synthesis, Herbicidal Activity, Crop Safety and Soil Degradation of Pyrimidine- and Triazine-Substituted Chlorsulfuron
Derivatives" is devoted to a very important topic. The development of safe agricultural chemicals is pivotal for all the humanity. The present manuscript provides a comprehensive research on a series of chlorsulfuron
derivatives as herbicides. In my opinion, the quality of this paper is very high and the importance of this research is very significant, and thus, the manuscript can be accepted in the present form.

1. "And compounds like W111 is potential green" should be "And compounds like W111 are potential green".

Reviewer 3 Report

This is the well-written manuscript in which the experimental data support the Authors' claims. I suggest to accept this manuscript for publication after following revisions:

1) Introduction section is too laconic and should be improved.

2) The Authors submitted the SI files as seperate files. I recommend to merge those files into one SI file.

3) I would recommend to move some Tables from the main ms file to SI to make the text more reader-friendly and to present the crucial data. The same is for the Figures, in Reviewer's opinion, such too large number of Tables and Figures in the main ms text, makes the text reader-unfriendly.

4) Subsection 3.1 -> all NMR spectra should be dicsussed (maybe in SI), mentioning an example is not enough.

5) Novelty of this work should be further highlighted, also including the previous works reported by the Authors.

6) NMR data is just noted, there is a need to present all the spectra in SI.

Round 2

Reviewer 1 Report

The authors have done a good job of improving the manuscript. I believe that in its present form the article can be accepted for publication in Molecules.